# Detection of Lumbar Spondylolisthesis from X-ray Images Using Deep Learning Network

**DOI:** 10.3390/jcm11185450

**Published:** 2022-09-16

**Authors:** Giam Minh Trinh, Hao-Chiang Shao, Kevin Li-Chun Hsieh, Ching-Yu Lee, Hsiao-Wei Liu, Chen-Wei Lai, Sen-Yi Chou, Pei-I Tsai, Kuan-Jen Chen, Fang-Chieh Chang, Meng-Huang Wu, Tsung-Jen Huang

**Affiliations:** 1International Graduate Program in Medicine, College of Medicine, Taipei Medical University, Taipei 11031, Taiwan; 2Department of Trauma-Orthopedics, College of Medicine, Pham Ngoc Thach Medical University, Ho Chi Minh City 700000, Vietnam; 3Department of Pediatric Orthopedics, Hospital for Traumatology and Orthopedics, Ho Chi Minh City 700000, Vietnam; 4Institute of Data Science and Information Computing, National Chung Hsing University, Taichung City 402, Taiwan; 5Department of Radiology, School of Medicine, College of Medicine, Taipei Medical University, Taipei 11031, Taiwan; 6Department of Medical Imaging, Taipei Medical University Hospital, Taipei 11031, Taiwan; 7Research Center of Translational Imaging, Taipei Medical University Hospital, Taipei 11031, Taiwan; 8Department of Orthopedics, School of Medicine, College of Medicine, Taipei Medical University, Taipei 11031, Taiwan; 9Department of Orthopedics, Taipei Medical University Hospital, Taipei 11031, Taiwan; 10Center for Measurement Standards, Industrial Technology Research Institute, Hsinchu 30044, Taiwan; 11Biomedical Technology and Device Research Laboratories, Industrial Technology Research Institute, Hsinchu 31057, Taiwan; 12TMU Biodesign Center, Taipei Medical University, Taipei 11031, Taiwan

**Keywords:** deep learning, LumbarNet, lumbar spine, spondylolisthesis, U-Net

## Abstract

Spondylolisthesis refers to the displacement of a vertebral body relative to the vertrabra below it, which can cause radicular symptoms, back pain or leg pain. It usually occurs in the lower lumbar spine, especially in women over the age of 60. The prevalence of spondylolisthesis is expected to rise as the global population ages, requiring prudent action to promptly identify it in clinical settings. The goal of this study was to develop a computer-aided diagnostic (CADx) algorithm, LumbarNet, and to evaluate the efficiency of this model in automatically detecting spondylolisthesis from lumbar X-ray images. Built upon U-Net, feature fusion module (FFM) and collaborating with (i) a P-grade, (ii) a piecewise slope detection (PSD) scheme, and (iii) a dynamic shift (DS), LumbarNet was able to analyze complex structural patterns on lumbar X-ray images, including true lateral, flexion, and extension lateral views. Our results showed that the model achieved a mean intersection over union (mIOU) value of 0.88 in vertebral region segmentation and an accuracy of 88.83% in vertebral slip detection. We conclude that LumbarNet outperformed U-Net, a commonly used method in medical image segmentation, and could serve as a reliable method to identify spondylolisthesis.

## 1. Introduction

The human spine supports the central axis of the body and comprises seven cervical, twelve thoracic, five lumbar, five sacral, and four coccygeal vertebrae. The lateral side of the spine presents a double S-bend, which forms a physiological curvature along the neck, chest, abdomen and pelvis to provide flexibility during movement [1,2]. Spinal degenerative diseases often appear at the lower parts of the lumbar (L4-S1) and cervical spine (C4-C7) since they are subject to the highest dynamic and static forces [2,3]. Spinal deformities due to spinal degeneration can compress or pull on the nerve roots, resulting in radicular symptoms, back pain or leg pain.

Spondylolisthesis is the displacement of a vertebral body relative to the vertebra below it [4,5,6]. Anterolisthesis and retrolisthesis refer to the forward and backward displacement, respectively, relative to the vertebra below [6,7]. Approximately 50%–81% of spondylolisthesis cases result from lumbar spondylolysis [4]. Spondylolisthesis has a variety of etiologies, such as hereditary factors, disc degeneration, and frequent engagement in activities involving repetitive lumbar hyperflexion and high rotation loads. Groups at high risk of developing spondylolisthesis include manual workers, weightlifters, gymnasts, dancers and football players [5,7]. Retrolisthesis has historically been considered an incidental finding that does not result in clinically significant symptoms. However, more recent studies have shown that its prevalence was in fact higher than otherwise suggested [8]. It can be observed in people with degenerative spinal conditions, post-spinal trauma, lumbar disc herniation or adjacent segment disease. The rate of retrolisthesis as detected from extension radiographs can reach up to 30% among people with chronic low back pain.

While computed tomography (CT), magnetic resonance imaging (MRI), and bone scans are useful for identifying the lesion site [5], physicians generally recommend lateral lumbar plain film radiographs in conjunction with flexion and extension views to assess spondylolisthesis and the degree of spinal instability. These specific radiographic views offer quick results, are inexpensive, and can be easily obtained in a clinic or primary hospital. The Wiltse–Winter classification distinguishes spondylolisthesis into six types by etiology (dysplastic, isthmic, degenerative, traumatic, pathologic, iatrogenic), with the most common being isthmic, degenerative, and dysplastic [5,7]. Spondylolisthesis, especially the dysplastic and degenerative types, can result in spinal canal stenosis, low back pain or leg pain, and neural compression symptoms [7,9]. To distinguish between true spondylolisthesis and pseudospondylolisthesis (the degenerative type), Bryk and Rosenkranz proposed the spinous process sign on X-ray images: in spondylolisthesis, the dorsal aspect of the spinous process reveals a step-off above the level of the vertebral slip; whereas, this sign presents below the level of vertebral slip in pseudospondylolisthesis [10].

Moreover, the Meyerding classification with five variants is widely used in clinical practice and research owing to its simplicity and ease of application [7,11]. The degree of slippage is classified into five grades by evaluating the extent to which the superior vertebra is displaced relative to the inferior vertebra. Specifically, the ratio of the bare displacement value of the upper vertebra (A) to its width (B) is calculated to obtain a P-grade (as illustrated in Figure 1). Table 1 outlines the relationship between the P-grade and the degree of vertebral slippage. With 50% as the threshold, a P-grade that is either less or more than 50% corresponds to mild or severe slippage, respectively. Of note, a P-grade greater than 100% represents a complete vertebral slippage, the most serious presentation of the condition [12].

The latest statistics from the AO (Arbeitsgemeinschaft für Osteosynthesefragen) Foundation revealed that the annual global incidence of degenerative spinal diseases and spondylolisthesis were 3.63% and 0.53%, respectively [13]. With an aging global population, spinal diseases are becoming increasingly prevalent, and are thus expected to place a greater strain on economies and healthcare systems. In particular, with the lack of resources and medical staff, physicians will be faced with an increased workload regarding the diagnosis or prognosis of spinal diseases such as spinal deformities [14,15,16], spondylolisthesis [17,18], spinal stenosis [19,20], herniated intervertebral disc (HIVD) [21,22,23], etc. While several therapeutics are available for spondylolisthesis, surgery remains the mainstay of treatment in patients where conservative management options have failed. However, it is still unclear whether the optimal surgical treatment is the use of decompression alone, or decompression in combination with either non-instrumented or instrumented fusion [24]. In particular, for people with co-existing spinal disorders, a comprehensive diagnosis and selection of favorable treatment represent challenges for physicians. Therefore, accurate and automatic solutions in detecting or formulating treatment plans for spinal diseases, including spondylolisthesis, are required.

Image segmentation is used to divide a digital image into multiple elements or regions and is widely applied in various fields, especially in medicine. The rapid development of machine learning in general and deep neural network in particular have accelerated advances in automated image recognition, with convolutional neural networks (CNNs) yielding considerable improvements in semantic segmentation [25,26,27,28]. Segmentation techniques are fundamental to object identification in medical images, such as those obtained using X-ray, MRI, and CT scan [21]. The main task of semantic segmentation is to identify the class labels in a pixel-by-pixel manner given the context of an image [25,29]. In medicine, this technique has typically been applied to images of cancer cells, muscle tissue, the aortic wall, and the skeletal system [30]. The implementation of deep-learning-based image segmentation, detection, and classification in lumbar spine radiographs, however, has been relatively scarce, due to the marked contrast of X-ray images and complexity of spinal structures [31].

Formulated to detect the vertebrae, the active shape model (ASM) segments X-ray images of the vertebrae by using edge polygon approximation [32]. Although the ASM exhibited favorable efficiency in experiments, its execution time and applicability in clinical settings remained areas of concern. Therefore, the authors proposed a parallel hybrid implementation, in which vertebral features are extracted with multiple graphics or central processing units (GPU or CPU) running in parallel to enhance performance.

Ronneberger et al. formulated U-Net, an end-to-end training network that contains an encoder–decoder network structure bearing a symmetrical U-shaped structure that uses paired primitives and markers during training. The model was then trained by U-Net to process a cellular image to output a binary map. U-Net is characterized by its superior training performance on data sets covering a small set of examples, achieving an accuracy of up to 92% in an experiment with medical images (512 × 512 in resolution) when executed using GPU acceleration [33].

Konya et al. trained a model on 730 X-ray images of the lateral lumbar spine by using deep neural networks. Their workflow, from start to finish, comprised data retrieval and preparation, model training and output, and analysis [34]. Various methods, including U-Net [33], Mask R-CNN [35,36], PSPNet [37], DeepLabV3 [38], and YOLACT [39], were implemented. Four indexes of accuracy were adopted, namely pixel accuracy average, mean intersection over union (IOU) average, mean accuracy average, and frequency weighted IOU average. Both U-Net and YOLACT exhibited favorable segmentation performance. The results demonstrated that these methods could support decision-making in future processing pipelines.

With regard to the measurement of critical values of the lumbar vertebrae, Cho et al. developed a machine learning- and computer vision-based method to quantify the extent of lumbar lordosis (LL) from radiological images. A total of 780 lateral X-ray images were used for training, testing, and data augmentation to improve contrast, and intensity normalization was performed to synthetically generate 12,580 images for training. Their U-Net segmentation achieved a dice score of 0.821 and a mean absolute error for LL of $8.055^{\circ }$ [40].

One limitation of the U-Net architecture is that it does not bear any fully connected layers. Furthermore, only the verification part of each convolution is used, which allows the segmentation map to contain each pixel of the input image in its full context. In addition, U-Net acts similar to masking two groups of segmented regions, with one being the foreground and the other being the background. To improve the accuracy of semantic segmentation, SegNet with a pixel-wise classification layer was proposed, which showed favorable efficiency since it saves the max-pooling indexes of feature maps in its decoder network. However, both efficiency and accuracy are key in semantic segmentation, and necessitate improvement in pixel-wise segmentation [41]. The feature fusion architecture was first used in a feature fusion single-shot multibox detector (FSSD). The feature fusion module (FFM) generates detailed feature maps in the object detection part [42]. A bilateral segmentation network bearing a two-path architecture composed of a spatial path and a context path, was constructed by Chen et al. [43], and Yu et al. [44], in which the spatial path could preserve the spatial information of original images. The network reached 105 frames per second (FPS) in the Cityscapes data set, which contained images with a resolution of 1024 × 2048. Specifically, we followed the description outlined by Yu et al. and termed our two blocks “multiply” and “add”. The “multiply” block was used to re-weight a feature map based on a feature-fusing map, while the “add” block added two feature tensors. This design is very similar to those used in most current and standard self-attention modules. Drawing on these works, we formulated a deep learning method that combines U-Net, FFM, and various image processing techniques to detect any abnormal slippages in lumbar X-ray images performed in the true lateral, flexion and extension lateral positions. To the best of our knowledge, this study is the first to construct a fully convolutional network for detecting spondylolisthesis from lumbar X-ray images. In particular, we determined the threshold values (K values) that signify the presence of vertebral slippage and evaluated the accuracy of our model. Based on U-Net and image measurement analysis, we hope to provide an automatic detection method of spondylolisthesis in X-ray images with high accuracy and low false-negative rate, and that this model will serve as a potential solution for healthcare systems in the near future.

Our study makes several primary contributions:This is the first study to address the problem of identifying lumbar slippage in true lateral, flexion and extension lateral views of X-ray images.LumbarNet presents a novel design for the detection of spondylolisthesis by using piecewise slope detection (PSD), dynamic shifting (DS), and the hybrid judgement of P-grade and PSD value to enhance slippage identification in the lumbar spine.Our method is robust against the presence of lumbar sacralization and reaches a high accuracy accordingly.

## 2. Materials and Methods

### 2.1. LumbarNet for Segmenting Vertebral Regions from X-ray Images

To detect lumbar slippage, all lumbar vertebrae and the sacral region must first be accurately segmented on X-ray images. In traditional image processing, either the binarization method with a determined threshold or an automatic binary method, such as Otsu, is used. However, the ROIs obtained using the former method are greatly influenced by the distribution of grayscale values in the foreground and background. Due to the various contrasts of X-ray images, we formulated LumbarNet, a deep convolutional neural network, instead of using traditional computer vision methods.

LumbarNet can be distinguished from U-Net by its “feature fusion module (FFM)”. U-Net, used in medical image segmentation, consists of two convolutional networks paths, a contraction and an expansion path, alternatively known as the encoder and decoder, respectively. The contraction path functions to extract features from the input image, and the expansion path samples the extracted features and generates the segmented images. However, U-Net, triggered by binary-cross entropy loss, generally cannot produce stable segmentation results if the quality and characteristics of a testing sample are not consistent with those of the training data set. To improve the encoder’s efficiency, we added the feature fusion module (FFM) to the encoder path. The FFM consists of convolutional layers with a stride function, batch normalization and rectified linear unit (ReLU) activation functions, average pooling layers, add operators, and a multiply operator. It was designed to reorganize the latent feature extracted by the encoding path, so that a more representative feature for lumbar segmentation could be obtained. Figure 2 details the schema of LumbarNet, including the encoder types, decoder type, and FFM.

The input layer of LumbarNet has a size of 512 × 512 × 3. LumbarNet comprises two main paths: a spatial and a downsampling path. The spatial path, preserving the spatial information and generating high-resolution features, contains three layers. Each convolutional layer has stride = 2, followed by batch normalization and Rectified Linear Unit (ReLU). It encodes most of the rich and detailed messages in the low levels, while the robust features extracted from the downsampling path are in the high levels.

FFM fuses these two feature paths at different levels. Firstly, the outputs of the spatial and the downsampling paths are concatenated through the features from different levels. Then, the scales of the features retain the balance by using the batch normalization. Next, a global pool is used for the concatenated features to form a feature tensor, and the weight is computed by a sigmoid function. This weight tensor can re-weight the features by multiplying the concatenated features and adding to the feature selection.

The same number of channels in the encoder and decoder is maintained by adding skip connections between the corresponding blocks in the downsampling and upsampling paths. The skip connections are used explicitly to copy features from the earlier layers to the subsequent layers. Nodes from the shallow layer and deep layer are concatenated through these skip connections. The deeper layer is then treated as a wider layer and is connected to the next layer. We used a dropout layer between these two convolutional layers to prevent overfitting and co-adaptation of the features. There are a total of 29 convolutional layers in the network. In the final layer, a 1 × 1 convolution, five-dimensional space, and a sigmoid activation function are used to output the probability map of the semantic segmentation, bearing the same size as the original 512 × 512 input. The detailed network structure with the parameter settings is illustrated in Figure 3.

LumbarNet uses a pixel-wise softmax function to calculate the loss function, and the cross entropy loss is defined as follows:(1)$$E=-\sum_{i=1}^{n}w(x)logp(x).$$

By classifying each pixel as an end-to-end image for learning and output, the inference and image processing operations of LumbarNet are executed according to the following procedure to confirm whether a patient has spondylolisthesis or not.

### 2.2. Detection of Lumbar Spondylolisthesis

The original image was resized to 512 × 512 pixels. Next, LumbarNet was used to generate the output image. Nearest-neighbor interpolation was then applied to scale the segmented image to its original size. The segmented image has three labels, namely the background, vertebral regions, and the sacrum. Thereafter, ROIs on the labeled image could be extracted using the contour finding method. Through the use of eight connected components, each vertebral and sacrum region can be detected. Finally, the fitted quadrilateral of the vertebral region is calculated using a pole.

#### 2.2.1. P-Grade

There are five lumbar vertebrae in total. The quadrilateral of each vertebra $L_{i}$ has four extreme points, where i∈{1,2,3,4,5}. The upper-left, upper-right, lower-right, and lower-left points of a lumbar vertebra $L_{i}$ are $p_{L_{i,1}}$, $p_{L_{i,2}}$, $p_{L_{i,3}}$, $p_{L_{i,4}}$, respectively. The sequence of the four points of each vertebra runs counterclockwise from the upper-left point. In addition, the top plate of the sacrum is represented by two points, namely the left $p_{sacral_{1}}$ and right $p_{sacral_{2}}$ points, as illustrated in Figure 4.

After these feature points are detected, a P-grade for the difference between the left and right vertebral segments is calculated, with a value >20% indicating abnormality. The process of determining abnormality from one X-ray image is illustrated in Figure 5.

Each P-grade corresponding to the L1-L2, L2-L3, L3-L4, L4-L5, and L5-S1 levels is calculated, and the algorithm checks whether it is greater than K1, as indicated in the following equations:(2)$$f_{pgrade}\left(m,n\right)=p_{grade}\left(i,j\right)>K1,$$
(3)$$p_{grade}\left(i,j\right)=\frac{D(P_{i,proj},P_{j,2})}{\sqrt{∥{P_{j,1}-P_{j,2}∥}^{2}}},$$
(4)$$D(P_{i,proj},P_{j,2})=\sqrt{∥P_{i,proj}-p_{j,2}{∥}^{2}}$$
where $f_{pgrade}$ is the shift value of the lower plate of the *i*-th lumbar vertebra relative to the upper plate of the *j*-th lumbar vertebra. Both *i* and *j* are indexes of the proximal vertebral regions, specifically i∈{L1,L2,L3,L4,L5} and j∈{L2,L3,L4,L5,S1}. Subsequently, the algorithm determines the projected point $p_{proj}$ between the point $p_{i,3}$ and line $L_{j}$, the end points of which are $p_{j,1}$ and $p_{j,2}$. Thereafter, the distance $D_{p_{i,3},L_{j}}$ between the projected point and upper right point, $p_{j,2}$, of the *j*-th lumbar vertebra is computed. In addition, $f_{pgrade}$ is computed by dividing $D(P_{i,proj},P_{j,2})$ by the upper-plate length of the *j*-th lumbar vertebra, as illustrated in Figure 6. The projected point is either within or beyond the segmented line, which marks the upper plate of the *j*-th lumbar vertebra.

#### 2.2.2. Piecewise Slope Detection (PSD)

As mentioned, each lumbar vertebra has four extreme points, $p_{L_{i,1}}$, $p_{L_{i,2}}$, $p_{L_{i,3}}$, and $p_{L_{i,4}}$. The vertical shift angle is measured and used to directly detect the variation in angle between the proximal lumbar region and its gap. The angle θ is calculated using the cosine theorem for the gap between the upper and lower points and the corresponding vertical line. Because slippage of the lumbar vertebrae is present in anterolisthesis and retrolisthesis, both the left and right sides of the lumbar vertebrae are considered in the calculation of the degree. Specifically, the angles $\theta_{\alpha }$ and $\theta_{\beta }$ correspond to the left and right sides of each vertebra, respectively. The subscripts α and β correspond to the difference in the slope for the left and right vertebral segments, respectively. The PSD method detects slippage using the following equations:(5)$$f_{slippage}\left(m,n\right)=\left|\theta_{i,j}-\theta_{i,j+1}\right|>K2,$$
(6)$$\theta_{\left(i,j\right)}=\cos^{-1}\frac{\vec{U}\cdot \vec{V}}{∥U∥\times ∥V∥},$$
where $i\in \left\{\alpha ,\beta \right\}$, j∈{1,2,...,10}, and (m,n) are for the proximal vertebrae from $L_{1}$ to $S_{1}$ and are in adjacent paired regions. If (m,n) is equal to $\left(L_{1},L_{2}\right)$, then slippage might be present between the first and second lumbar vertebrae for any *j* = 1, 2, and 3. If (m,n) is equal to $\left(L_{5},S_{1}\right)$, then slippage might be present between the fifth lumbar vertebra and sacrum for any *j* = 9 and 10. Each θ is computed using Equation (Equation 6), where *U* is a vector for two consecutive points on a segment and *V* represents the vertical points. Each point and segment of the first and second lumbar vertebrae are detailed in Figure 7.

#### 2.2.3. Dynamic Shift (DS) Detection

DS detection is applied to two X-ray images in the flexion and extension positions to calculate the shift in the proximal lumbar region when the patient leans forward and backward, respectively. The procedure for DS detection is illustrated in Figure 8.

Similarly, Equation (Equation 4) is used to calculate the shift in the proximal lumbar region and $f_{ds}$ is used to check whether the shift distance is greater than K3. The formula is as follows:(7)$$f_{ds}\left(i,j\right)=\left|D(P_{i,proj},P_{j,2})-D^{{}^{\prime }}\left(P_{i,proj}^{{}^{\prime }},P_{j,2}^{{}^{\prime }}\right)\right|>K3,$$
where $D(P_{i,proj},P_{j,2})$ and $D^{{}^{\prime }}\left(P_{i,proj}^{{}^{\prime }},P_{j,2}^{{}^{\prime }}\right)$ are the shift distances when leaning forward and backward, respectively, each determined from separate X-ray images as illustrated in Figure 9. The index $i\in \left\{L1,L2,L3,L4,L5\right\}$, and the index $j\in \left\{L2,L3,L4,L5,S1\right\}$. If only one X-ray image of the patient is available, the P-grade and PSD methods are executed. However, if the patient has more than two X-ray images, the P-grade, PSD, and DS methods are executed, in this order, to identify spondylolisthesis.

The *K* values (K1,K2,K3) were determined from 394 cases through the following steps: Firstly, two spinal surgeons reviewed these cases and recorded whether they found signs of spondylolisthesis on the X-ray images. Then, we obtained the values by using P-grade, PSD and Dynamic Shift algorithm. Finally, we calculated the golden values, named K1,K2,K3 representing P-grade, PSD, Dynamic Shift, respectively. Using the golden values as criteria, we could acquire the best accuracy. Coding, known as Macro in Microsoft Excel, was used to calculate the golden values (K1,K2,K3). We also wrote the formula on Microsoft Excel to calculate the accuracy, sensitivity, and 95% confidence interval.

## 3. Results

### 3.1. Data Set

Lumbar X-ray images were captured with a Radnext 50 X-ray machine, Hitachi Global, Tokyo, Japan. Patients were required to either stand erect or lean forward and backward. After a full IRB approval was granted (TMU-Joint Institutional Review Board No. N201807084), 706 X-ray images of patients with low back pain were collected at Taipei Medical University Hospital. While the images were taken under the same settings (including the same X-ray energy), the digital images differed greatly in terms of contrast and grayscale range. The vertebral and sacral regions in these images were then manually labeled. The extent of slippage in each X-ray image was scored by two orthopedists.

Overall, we obtained 312 X-ray images of lumbar vertebrae with abnormalities, among which 250 images formed our training subset and the remaining 62 images formed our testing subset. All patients’ personal information were removed. We expanded our data set through data augmentation using transformations such as rotation, changes in contrast, and flipping. The images ranged widely in resolution from 494 to 2456 pixels in width and from 888 to 3408 pixels in height.

In the training model, the learning rate was $1\times 10^{-4}$, the Adam optimizer was used, the training period was 200 epochs, and the loss function was calculated as per Equation (Equation 1). The training accuracy was determined for each epoch, and the loss function is illustrated in Figure 10. Training and evaluation took 16 h and 30 ms, respectively, using a computer with an NVIDIA GTX 1080 Ti GPU. The accuracy reached 99.49% during the training phase. We then verified whether the learned geometric region intersected the artificially labeled ROIs by applying the learned model to the testing set to calculate mean intersection over union (mIOU). A true positive was indicated by a rectangular IOU probability value greater than 0.5. Ultimately, the mIOU of U-Net was 0.8 and the mIOU values for L1,L2,L3,L4,L5, and the sacral regions were 0.76, 0.89, 0.83, 0.75, 0.76 and 0.83, respectively. For LumbarNet, the mIOU was 0.88 and the mIOU values for L1,L2,L3,L4,L5, and the sacral regions were 0.852, 0.889, 0.889, 0.889, 0.889 and 0.889, respectively (Table 2). Comparison of the performance between LumbarNet and U-Net, shown in Table 2, is an ablation study for verifying the effectiveness of our feature fusion module (FFM). In addition, because we only adopted cross-entropy loss in our design, no ablation study for loss terms was required.

### 3.2. Analytical Experiments

#### 3.2.1. K1 Experiments for P-Grade

The relationship between the bare displacement value A of the lower vertebra and the width B of the upper vertebra is calculated as follows:(8)Pgrade=A/B×100

The extent of vertebral slippage between the projected point and the upper-right point ($p_{j,2}$) of the *j*-th lumbar vertebra was calculated. The shifted identification algorithm was applied to 394 cases for training, and a threshold value of K1 = 10 was suggested during training. The accuracy, sensitivity, specificity, false-positive rate, and false-negative rate were 88.05% (95% CI, 80%–91%), 89.44% (95% CI, 79%–94%), 84.92% (95% CI, 81%–86%), 7.32% (95% CI, 4%–6%) and 4.63% (95% CI, 4%–15%), respectively (illustrated in Table 3).

#### 3.2.2. K2 Experiments for PSD

The *K* value in the vertebral anomaly test algorithm was tested according to the samples and the better value of *K* was analyzed. In general, a larger *K* value indicates a greater “allowable displacement” between the upper and lower vertebrae. The *K* value was tested in 100 test points in sequence, and the model performance was determined in terms of the precision and false-positive rate. The test points were obtained from all images, and the model results were tested against the judgment of orthopedists with regard to the location of vertebral slippage. A higher and lower *K* value correspond to a smaller and larger allowance, respectively, with respect to the accuracy and false-positive rate. A *K* value of 37 was selected as the parameter of subsequent slip abnormality detection. The accuracy, sensitivity, specificity, false-positive rate, and false-negative rate were 81.22% (95% CI, 75%–90%), 85.82% (95% CI, 81%–92%), 71.43% (95% CI, 61%–90%), 9.14% (95% CI, 4%–12%) and 9.64% (95% CI, 5%–13%), respectively (illustrated in Table 3).

#### 3.2.3. K3 Experiments for DS

The DS method was applied to X-ray images of the patients leaning forward and backward. If the difference between the shift distances for the forward and backward motions was greater than K3, spondylolisthesis was indicated. The final value of K3 was 0.14. Our trained model exhibited an accuracy, sensitivity, specificity, false-positive rate and false-negative rate of 81.42% (95% CI, 80%–82%), 82.5% (95% CI, 79%–83%), 80.31% (95% CI, 79%–81%), 9.67% (95% CI, 9%–10%) and 8.91% (95% CI, 8%–10%), respectively (illustrated in Table 3).

The thresholds of *K* values are indicated in Figure 11. Furthermore, the combination of P-grade (K1 = 10) and PSD (K2 = 50) methods yielded the optimal results (Table 3), as indicated by an accuracy of 88.83% (95% CI, 84%–91%), sensitivity of 91.24% (95% CI, 85%–93%), specificity of 83.33% (95% CI, 81%–84%), false-positive rate of 5.08% (95% CI, 4%–6%), and false-negative rate of 6.09% (95% CI, 5%–11%).

## 4. Discussion

To the best of our knowledge, there has not been any algorithm designed for the detection of spondylolisthesis in lumbar X-ray images via the use of P-grade, PSD and DS. Our method was developed to detect exactly where the spondylolisthesis would occur, and to assess the relative displacement of two adjacent vertebrae. Hence, motivated by clinical requirements, we consider our method to be highly specialized for spondylolisthesis detection. This provides the reasoning for why we compared our LumbarNet with U-Net only.

Our results are similar to those of previous studies applying machine learning to detect spinal pathologies. Specifically, the faster adversarial recognition (FAR) network of Zhao et al. for grading spondylolisthesis exhibited high accuracy (0.9883±0.0094 in training, 0.8933±0.0276 in testing) when used for the detection of vertebral abnormalities in the absence of landmarks [45]. Furthermore, Ansari et al. reported a high accuracy of 92.05% for their generalized regression network, which outperformed a neural network and support vector machine (SVM); their method was used to classify suspected spinal pathologies into healthy spine, disk hernia, and spondylolisthesis [46]. In addition, Karabulut et al. reported a high accuracy of 89.73% for their synthetic minority oversampling technique (SMOTE), which was used for automatic classification and to assist in delivering the appropriate management for patients [47]. Finally, Akben et al. reported a success rate of 84.5% for their naive Bayes classifier when used with a combination of three or more attributes. They also revealed a 96.13% accuracy rate for their method in the detection of spondylolisthesis; however, this rate was considered to be misleading due to asymmetry in the size of their samples [48]. Nonetheless, our accuracy rates are lower than those reported by Unal et al. By combining a fuzzy C-means (FCM) algorithm with a naive Bayes classifier or SVM, they achieved high accuracy rates of 97.42% and 96.45%, which were substantially higher than the 82% to 85% rates achieved using the SVM [49,50].

Our study had several limitations. First, while two orthopedists independently reviewed and confirmed spondylolisthesis in the X-ray images, the effects of interobserver and intraobserver variability were not evaluated. Second, our model did not specify the etiology and the slippage grade of spondylolisthesis since our goal was to propose a computer-aided diagnostic algorithm for automatically detecting vertebral slippage in X-ray images. Furthermore, the spinal anatomy is highly complex with multiple interconnected and overlapping structures, especially in the X-ray images of patients with spinal degenerative diseases. For instance, the presence of osteoporosis will affect the results of segmentation owing to the poor image quality (as illustrated in Figure 12a). In addition, it is more difficult to identify the fifth lumbar vertebra (L5) and the first sacrum (S1) compared with the other lumbar vertebrae owing to its superimposition with the pelvis or the presence of lumbar sacralization, a variant of L5 and S1 (as illustrated in Figure 12b). In addition, vertebral osteophytes (bone spurs) will introduce errors into the quadrilateral fitting since they do not represent the true corners of vertebrae (as illustrated in Figure 12c). If the corners cannot be correctly detected, all assessments will surely fail. Therefore, lesion identification is more challenging in the X-ray images of patients with co-existing osteoporosis, degenerative spine disease, or lumbar sacralization.

Although these factors compromised the performance of our model, we believe the overall results warrant adequate consideration (illustrated in Figure 13). Thus, we hope our algorithm will be applied and further developed to detect, not only spondylolisthesis, but also other spinal diseases. In regions with limited resources and a shortage of healthcare professionals, this model can serve as a potential solution in detecting spondylolisthesis from easily obtained X-ray images.

## 5. Conclusions

To the best of our knowledge, LumbarNet is the first algorithm to employ P-grade, PSD, and DS to detect spondylolisthesis in true lateral, flexion and extension lateral views of lumbar X-ray images. Our model achieved an mIOU of up to 0.88 in determining the affected vertebral region. The ROI was then obtained to determine the end points, and slippage in the lumbar vertebrae was detected using the P-grade and PSD methods, achieving an accuracy rate of 88.83%. We believe that in the near future, our model can be applied as a potential solution for healthcare systems and further developed to detect a wider range of spinal disorders.

## Figures and Tables

**Figure 1 jcm-11-05450-f001:**
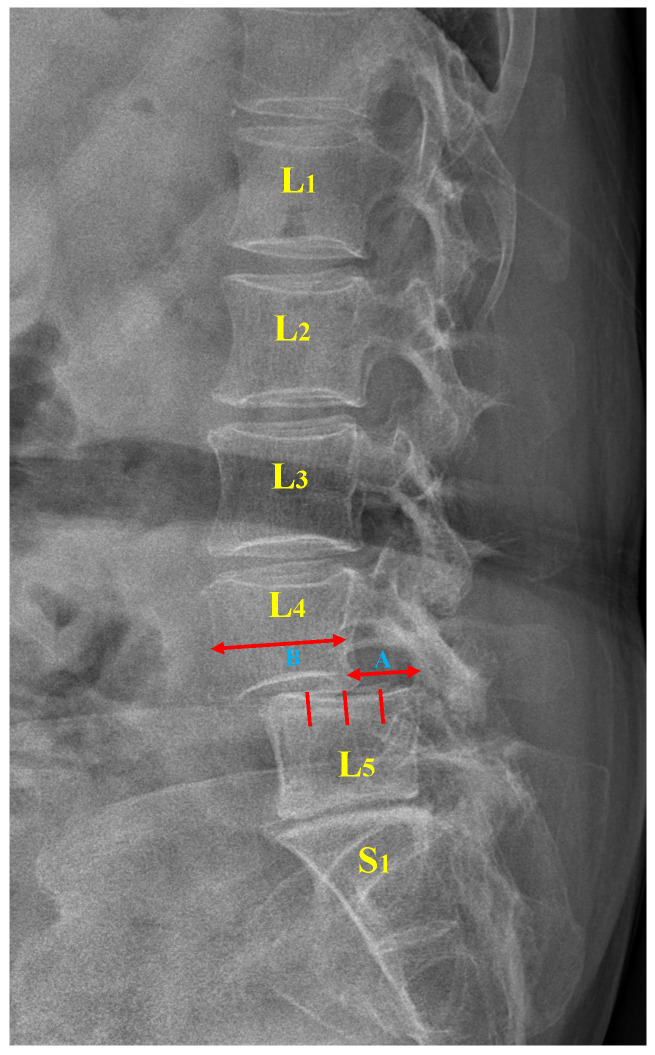
Lumbar spondylolisthesis. L: lumbar vertebra, S: sacral vertebra, A: slip distance between two vertebrae, B: width of the superior vertebra.

**Figure 2 jcm-11-05450-f002:**
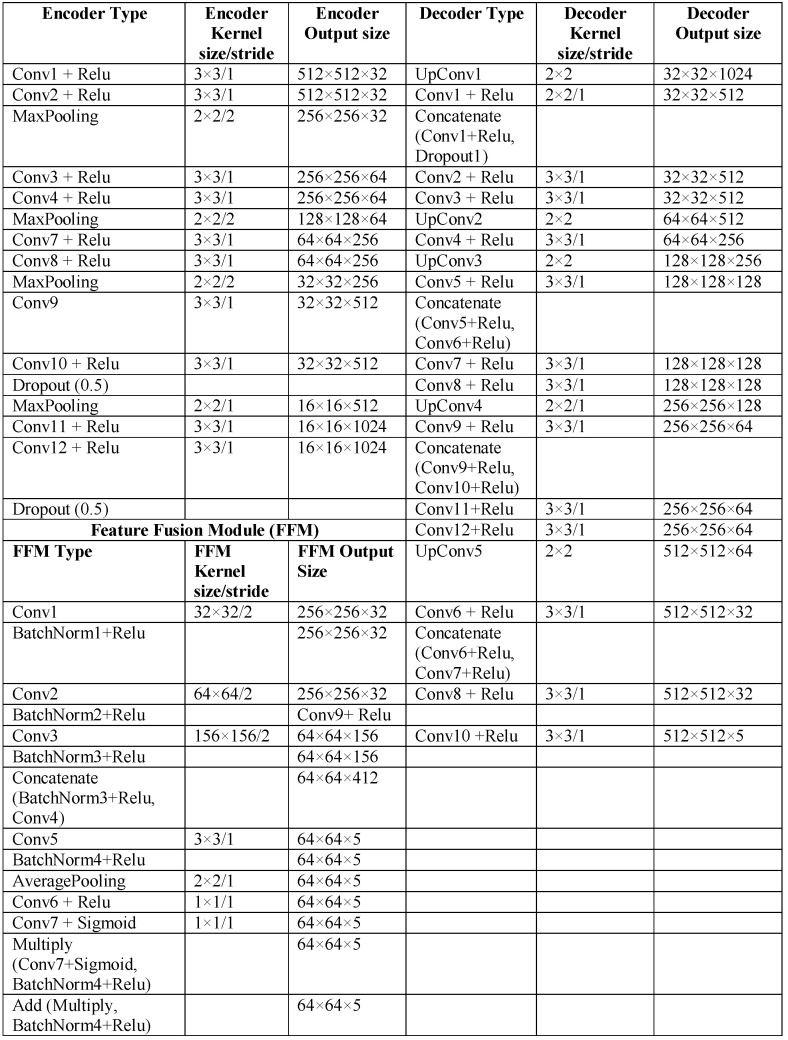
Architecture of LumbarNet.

**Figure 3 jcm-11-05450-f003:**
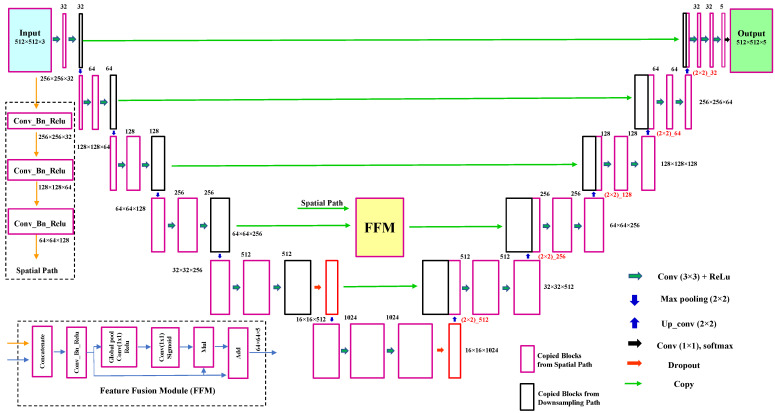
LumbarNet uses an end-to-end architecture for object semantic segmentation. It takes a three-channel input image at a resolution of 512 × 512 and produces an output containing five channels.

**Figure 4 jcm-11-05450-f004:**
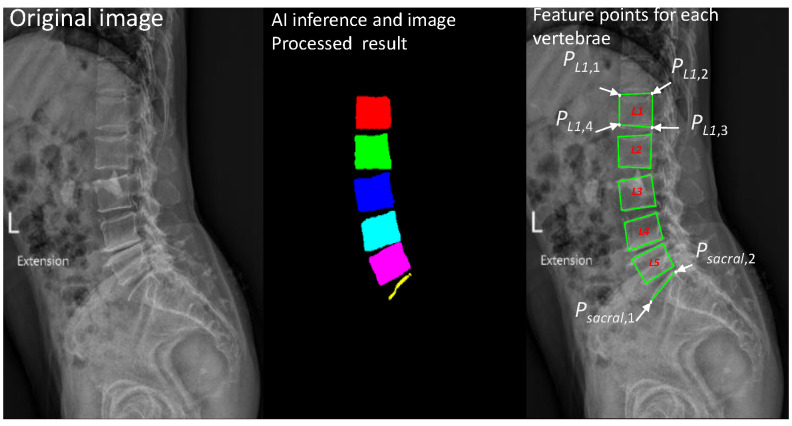
Original image (**left**), processed image (**middle**) with different colors to distinguish vertebrae from L1 to S1, and calculated feature points of each vertebra and sacrum (**right**).

**Figure 5 jcm-11-05450-f005:**
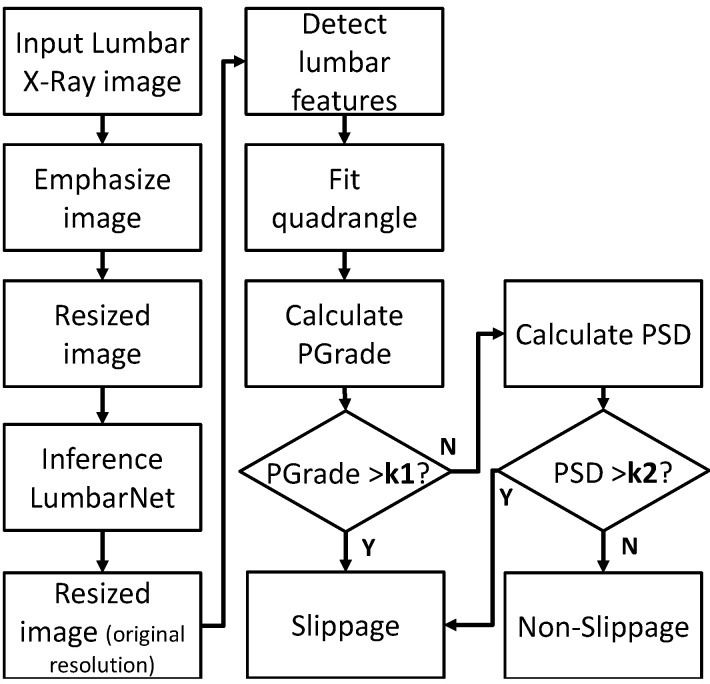
Process of detecting abnormal vertebral slippage in X-ray images by using P-grade.

**Figure 6 jcm-11-05450-f006:**
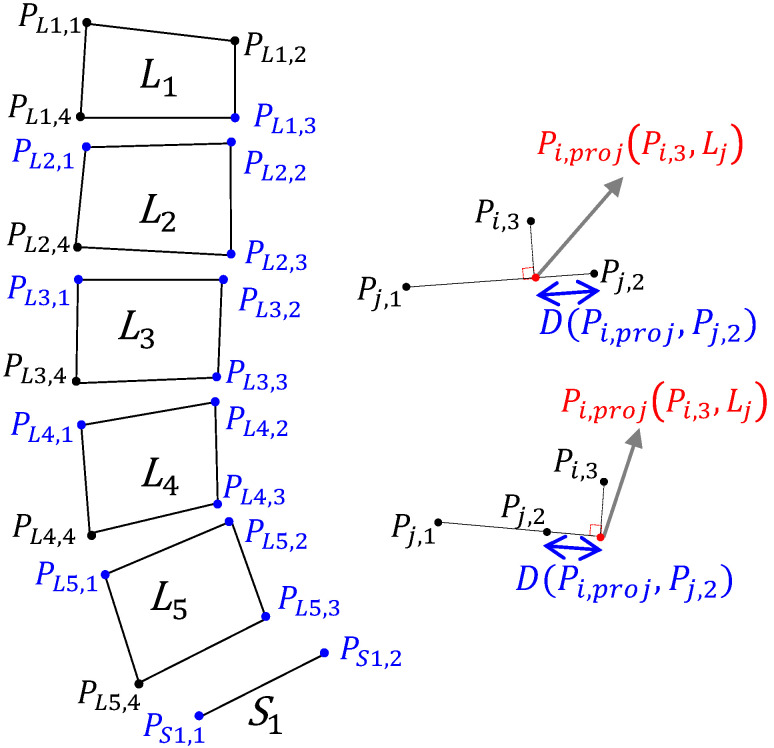
Five lumbar vertebrae with four points and the sacrum with two points. The projected point may lie either within or beyond the segmented line.

**Figure 7 jcm-11-05450-f007:**
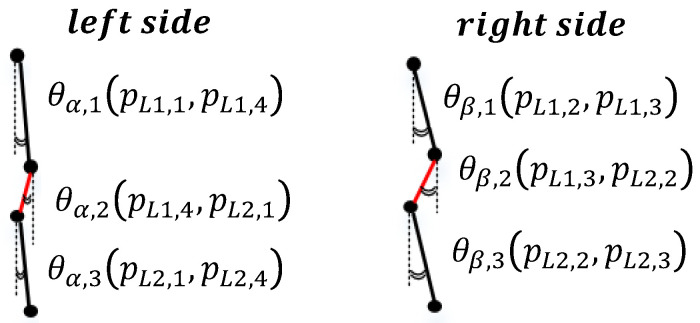
Descriptions of each angle in the PSD methodology. There are four extreme points of each lumbar vertebra, $p_{Li,1}$, $p_{Li,2}$, $p_{Li,3}$, and $p_{Li,4}$, which correspond to the upper-left, upper-right, lower-right, and lower-left points, respectively, of a lumbar vertebra *i*. Each vertebral region can be calculated from the left- and right-sided angles between the proximal line segments. $\theta_{\alpha }$ and $\theta_{\beta }$ represent the left and right side, respectively.

**Figure 8 jcm-11-05450-f008:**
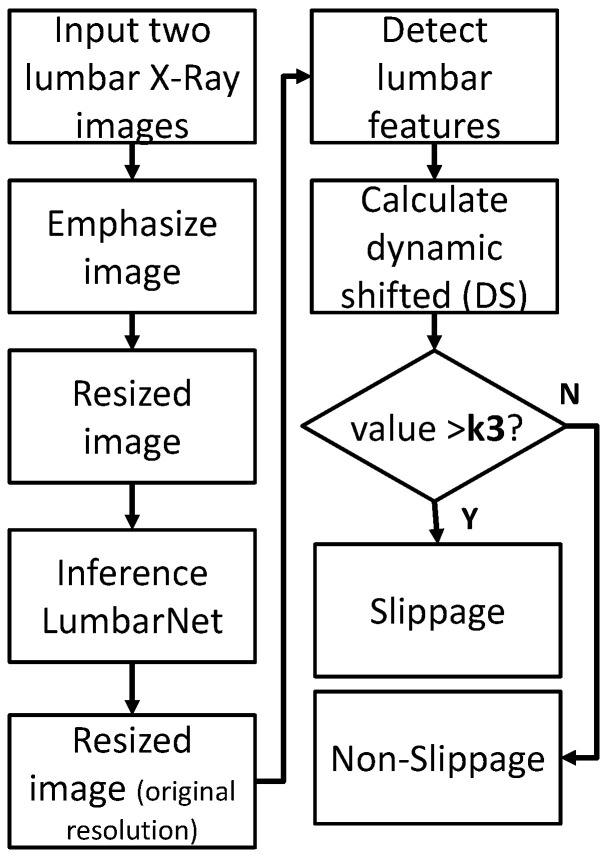
Process of detecting abnormal vertebral slippage in flexion and extension X-ray images by using dynamic shift detection.

**Figure 9 jcm-11-05450-f009:**
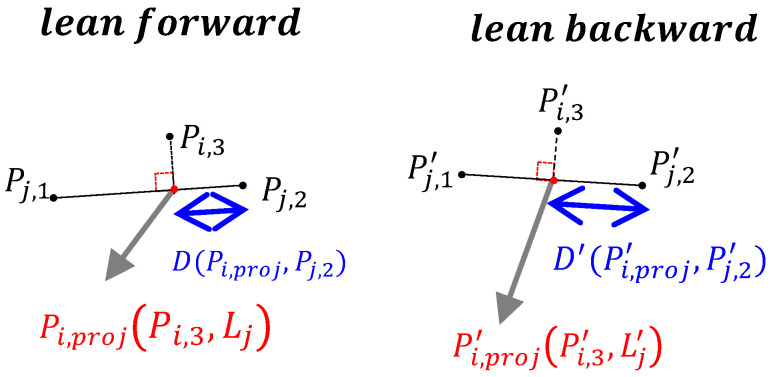
Dynamic shift for slippage detection.

**Figure 10 jcm-11-05450-f010:**
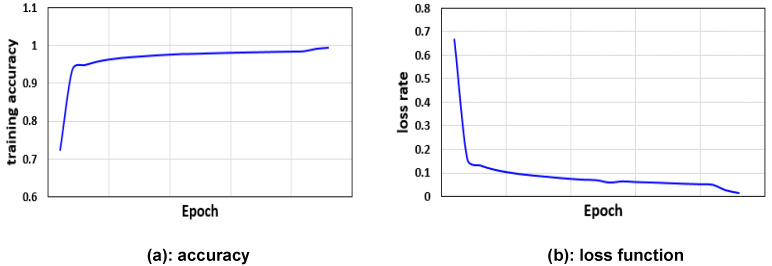
Training history of LumbarNet.

**Figure 11 jcm-11-05450-f011:**
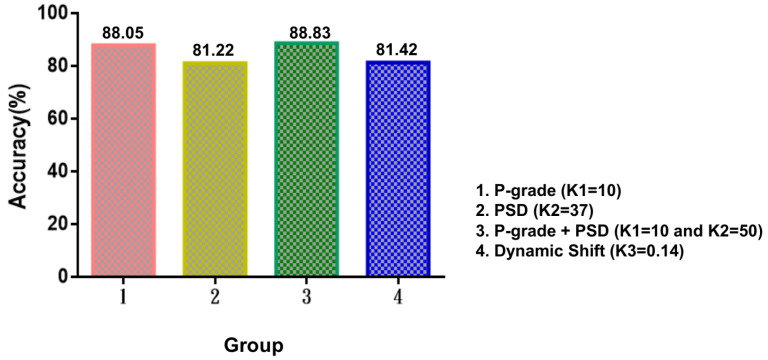
Accuracy of the algorithm in different conditions: P-grade, Piecewise slope detection (PSD), Dynamic shift (DS).

**Figure 12 jcm-11-05450-f012:**
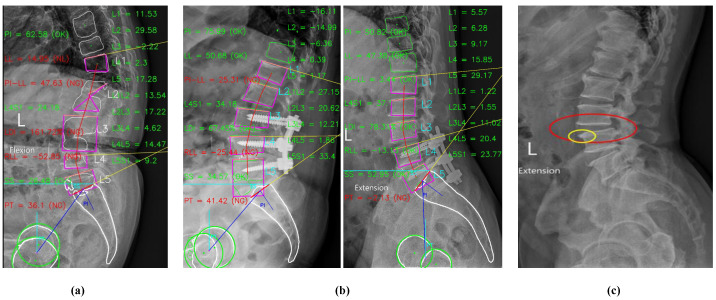
Errors in processing due to (**a**) osteoporosis makes poor image quality, (**b**) superimposition between the pelvis and vertebrae, and (**c**) vertebral spurs.

**Figure 13 jcm-11-05450-f013:**
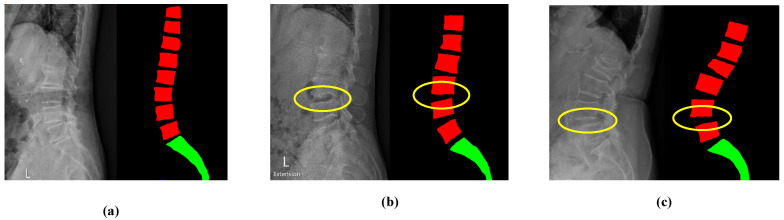
Performance of LumbarNet in the detection of lumbar spondylolisthesis: (**a**) No Spondylolisthesis, (**b**) Spondylolisthesis at *L*3-*L*4, (**c**) Spondylolisthesis at *L*4-*L*5.

**Table 1 jcm-11-05450-t001:** Classification of the vertebral slippage degree.

P-Grade (A/B)	%	Description
Grade 1	0–25%	Low grade
Grade 2	26–50%	
Grade 3	51–75%	High grade
Grade 4	76–100%	
Grade 5	Complete dislocation of vertebral body (>100%)	Spondyloptosis

**Table 2 jcm-11-05450-t002:** mIOU Comparison of U-Net and LumbarNet.

${\mathrm{mIOU}}_{0.5}$	L1	L2	L3	L4	L5	Sacrum	Average
U-Net	0.76	0.89	0.83	0.75	0.76	0.83	0.8
LumbarNet	0.852	0.889	0.889	0.889	0.889	0.889	0.88

**Table 3 jcm-11-05450-t003:** Accuracy, sensitivity, specificity, false-positive rate, and false-negative rate for spondylolisthesis detection.

	P-Grade	PSD	P-Grade + PSD	DS
	(K1 = 10)	(K2 = 37)	(K1 = 10 + K2 = 50)	(K3 = 0.14)
Accuracy	88.05%	81.22%	88.83%	81.42%
	(95% CI, 80%–91%)	(95% CI, 75%–90%)	(95% CI, 84%–91%)	(95% CI, 80%–82%)
Sensitivity	89.44%	85.82%	91.24%	82.50%
	(95% CI, 79%–94%)	(95% CI, 81%–92%)	(95% CI, 85%–93%)	(95% CI, 79%–83%)
Specificity	84.92%	71.43%	83.33%	80.31%
	(95% CI, 81%–86%)	(95% CI, 61%–90%)	(95% CI, 81%–84%)	(95% CI, 79%–81%)
False positive	4.63%	9.14%	5.08%	7.67%
	(95% CI, 4%–6%)	(95% CI, 4%–12%)	(95% CI, 4%–6%)	(95% CI, 9%–10%)
False negative	7.32%	9.64%	6.09%	8.91%
	(95% CI, 4%–15%)	(95% CI, 5%–13%)	(95% CI, 5%–11%)	(95% CI, 8%–10%)

## Data Availability

The data used for this study belong to the Industrial Technology Research Institute (ITRI) and Taipei Medical University Hospital (TMUH). We can upload a part of our numerical data at https://github.com/GIAM-TM/Detection-of-Lumbar-Spondylolisthesis-from-X-ray-Images-Using-Deep-Learning-Network.git (accessed on 7 August 2022). Part of the data can be found at these organizations.

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
