# Peer review of "Detection of Lumbar Spondylolisthesis from X-ray Images Using Deep Learning Network"

_jcm, 2022, doi:10.3390/jcm11185450_

Round 1
Reviewer 1 Report
Degenerative spondylolisthesis more than the image diagnosis is clinical condition affecting the quality of life of our patients in many cases specially in aging society, however the only radiological finding does not guarantee the clinical finding.
In my opinion it would have been more interesting any relationship with the patient clinical status.
Author Response
Dear Sir/ Madam,
Thank you very much for your comments and suggestions in our study.
May we give you the revised manuscript and responses to your comments? The deleted parts were all highlighted with Red color and the rewritten parts were all highlighted with Blue color in the text. Could you please see the attachment?
Thank you very much for your consideration.

Reviewer 2 Report
Comments on Detection of Lumbar Spondylolisthesis From X-Ray Images
Using Deep Learning Network
Congratulations tot he authors on establishing this neural-network-based spondylolisthesis detection and classification application, the material & methods section as well as the results section are well written, all the other sections need revision – find my comments below. Altogether this study is an interesting approach, nevertheless the accuracy – considering it is a rather simple task to identify spondylolisthesis – is not sufficient, furthermore in my humble opinion this application is not necessarily useful for clinical purposes but rather for research purposes, which the athors may consider in their discussion. Overall I believe that the manuscript can be considered after major revisions.
P1, line1-3 rephrase
P1, line 4 „more prevalent in …“ than what? – compared tot he rest of the population
General comment regarding the abstract: by what means are you addressing the problem of increasing prevalence of deg. Spondylolisthesis via a software to detect it? – the abstract as a whole needs tob e rewritten and aims and goals need tob e clarified.Results?
P2, line 22ff rephrase, also 24ff is somewhat incoherent and seems misplaced
P2, line 30ff Differentiate between spondylolisthesis vera and pseudospondylolisthesis!
P2, line38 not true
Figure 1 use xray not 6 squares to give an example – if needed at all
Include section 2 into the introduction – „related works“ is inadequate, neither should it be named lit. Review or whatsover bc it’s rather superficial
P5, line 171 rephrase
P5, line 177 typo
P12 line 308ff Set up a table or rephrase to give the reader a better overview
P12, line 337 It’s not clinical it’s radiological spondyloslisthesis you’re detecting
P13, line 362ff rephrase – not readable
Regarding conlusion – please rephrase the whle paragraph
Author Response

(The authors gave the same response as above.)

Round 2
Reviewer 2 Report
“and was proposed as a more accurate method to identify spondylolisthesis.” Rephrase, please consider having a native speaker look through the manuscript.
P2, line 56ff – if you refer to studies, cite them!
P3, line 83ff – if you think this sentence through, you may find that there is not much sense to it - if you operate with the indication “spondylolisthesis”, you should have detected anyway?!
P3, line 94 to reduce the mistake for physicians and give the 94 better treatment solutions for patients. Please simply delete this
P16, line 497ff rephrase, see above
P17, line 520 rephrase see above
Regarding your answer on the vera/falsa topic – I am well aware of which is which- very interesting summary though… nevertheless if you are trying to establish a model for spondylolisthesis diagnosis, you at least have to mention this aspect in your introduction!!!
The manuscript has been improved, nevertheless there are still some pitfalls, the use of the English language is partly still inadequate. Altogether I’d recommend another round of major revisions
Author Response
Dear Sir/ Madam,
Thank you very much for your comments and suggestions on the second round of peer review.
May we give you the revised manuscript and responses to your comments? The deleted parts were all highlighted with Red color and the rewritten parts were all highlighted with Blue color in the text. Could you please see the attachment?
Thank you very much for your consideration.
Best regards,
Meng-Huang Wu, M.D, Ph.D

Round 3
Reviewer 2 Report
After the 3rd round of review the quality of this manuscript has been improved significantly and my comments were adressed adequately. Congratulations on this study.